# TNF signaling mediates cellular immune function and promotes malaria parasite killing in the mosquito *Anopheles gambiae*

George-Rafael Samantsidis[1☯], Hyeogsun Kwon[1☯], Megan Wendland[1], Catherine Fonder[2], Ryan C. Smith[1]*

**1** Department of Plant Pathology, Entomology and Microbiology, Iowa State University, Ames, Iowa, United States of America, **2** Molecular, Cellular and Developmental Biology Interdepartmental Graduate Program, Iowa State University, Ames, Iowa, United States of America

☯ These authors are contributed equally.

\* smithr@iastate.edu

## Abstract

Tumor Necrosis Factor-α (TNF-α) is a proinflammatory cytokine and a master regulator of immune cell function in vertebrates. While previous studies have implicated TNF signaling in invertebrate immunity, the roles of TNF in mosquito innate immunity and vector competence have yet to be functionally explored. Herein, we confirm the identification of a conserved TNF-α pathway in *Anopheles gambiae* consisting of the TNF-α ligand, Eiger, and its cognate receptors Wengen and Grindelwald. Through gene expression analysis, RNAi, and *in vivo* injection of recombinant TNF-α, we provide direct evidence for the requirement of TNF signaling in regulating mosquito immune cell function by promoting granulocyte midgut attachment, increased granulocyte abundance, and oenocytoid rupture. Moreover, our data demonstrate that TNF signaling is an integral component of anti-*Plasmodium* immunity that limits malaria parasite survival. Together, our data support the existence of a highly conserved TNF signaling pathway in mosquitoes that mediates cellular immunity and influences *Plasmodium* infection outcomes, offering potential new approaches to interfere with malaria transmission by targeting the mosquito host.

## Author summary

Mosquito innate immunity is a major determinant of vector competence with significant implications in malaria transmission. During infection with the *Plasmodium* parasite, mosquitoes mount a sequence of immune signals originating from the mosquito midgut that stimulate the activation of mosquito immune cells (hemocytes) to limit parasite survival. Here, we provide compelling evidence that Tumor Necrosis Factor (TNF) signaling, a well-characterized immune pathway in vertebrates, is directly involved in mosquito hemocyte function and *Plasmodium*

Data availability statement: All relevant data are within the paper and its Supporting Information files.

Funding: This work was supported by R21AI144705 and R21AI166857 to RCS from the National Institutes of Health, National Institute of Allergy and Infectious Diseases. The funders had no role in study design, data collection and analysis, decision to publish, or preparation of the manuscript.

Competing interests: The authors have declared that no competing interests exist.

killing. We demonstrate that mosquito TNF signaling via the TNF ortholog, Eiger, requires the concerted function of the receptors Wengen and Grindelwald to control several aspects of mosquito immune cell biology and that ultimately limits malaria parasite survival. These data provide novel mechanistic insight into previously un-described roles of mosquito TNF signaling in anti-*Plasmodium* immunity that offer potential new molecular targets for malaria control.

## Introduction

*Anopheles* mosquitoes serve as the primary vectors of *Plasmodium* parasites, which cause malaria and impose substantial burdens on public health across the globe [1]. While there has been a significant reduction in malaria cases over the last twenty years due to improved vector control strategies, the continued effectiveness of these strategies has been jeopardized by increased insecticide resistance [2–4], which highlights the need to develop alternative approaches for malaria control. With recent advancements in gene-drive systems offering significant promise for population mod-ification [5–7], the potential that these genetic techniques can be used to manipulate the vector competence of mosquito populations to impair malaria transmission has become a reality. However, to fully leverage these genetic approaches, we require a better understanding of the molecular mechanisms that define *Plasmodium* infection in the mosquito host.

In response to *Plasmodium* infection, mosquitoes mount a series of sequential immune signals initiated by the midgut in response to ookinete invasion [8–12], which are further processed by the mosquito immune cells (hemocytes) [13–15] to promote malaria parasite killing [16–19]. As a result, hemocytes serve as integral immune mediators that directly contribute to ookinete recognition [16,17] or that promote humoral responses to limit oocyst survival [15,17,19]. Mosquito hemocytes have traditionally been classified into three main cell types based on morphological and biochemical properties: granulocytes, oenocytoids, and prohemocytes [20], with more recent single-cell studies expanding on the complexity of these cell populations [21,22]. Macrophage-like granulocytes are phagocytic and behave as immune senti-nels either in circulation in the hemolymph or as sessile cells attached to the midgut or other mosquito tissues [20,23]. Previous studies have demonstrated that granulocytes respond to stimuli resulting from ookinete midgut invasion to mediate both early- and late-phase immune responses against *Plasmodium* [15–18]. Oenocytoids have pri-marily been associated with the expression of prophenoloxidases (PPOs) [20], which are key enzymes in the melanization pathway and have been previously implicated in oocyst survival [19]. Lastly, prohemocytes are presumed precursors [20,24] that give rise to granulocyte and oenocytoid populations under infective conditions [13–15].

Mosquito hemocyte populations are highly heterogenic [21,22], with their composi-tion tightly regulated by a variety of signaling pathways in response to different phys-iological conditions. This includes previous studies that have demonstrated the ability of hemocytes to proliferate in response to blood-feeding [24,25], a process regulated

by the release of insulin-like peptides and subsequent activation of the PI3K/AKT and MAPK/ERK signaling pathways [26–29]. In addition, the Signal Transducer and Activator of Transcription (STAT) [14,15], the LPS-induced TNF-alpha factor (LITAF)-like transcription factor 3 (LL3) [15,30], c-Jun N-terminal kinase (JNK) [14], and Toll [14,30] pathways have been associated with hemocyte differentiation and parasite attrition. Furthermore, eicosanoid signaling pathways have been implicated in hemocyte function, differentiation, and *Plasmodium* killing [18,19,31,32], and are central to the establishment of innate immune memory that confers increased resistance to infection [18,31]. Yet, despite these advances, our understanding of the immune signals that modulate mosquito hemocytes remains limited.

Tumor Necrosis Factor-α (TNF-α) is one of the most important regulators of immune function in vertebrates, acting as a proinflammatory cytokine and critical mediator of immune cell regulation [33,34]. Across vertebrate systems, components of TNF signaling have remained conserved, consisting of a TNF-α ligand and two receptors: the TNFR1 and TNFR2 [34]. Previous phylogenetic analyses have revealed the presence of orthologous TNF pathways in invertebrates [35,36], however few studies have examined TNF signaling beyond *Drosophila*. In *Drosophila*, the TNF pathway is comprised of an analogous TNF ligand, *Eiger*, and its receptors, *Wengen* (*Wgn*) and *Grindelwald* (*Grnd*) [37]. Together, these TNF signaling components regulate multiple physiological processes in *Drosophila*, including tissue growth regulation, cellular proliferation, development, and host defense [38]. This includes significant contributions from hemocytes in mediating these physiological functions [39,40]. Under both homeostatic or infected conditions, Eiger modulates *Drosophila* hemocyte function to enhance survival through its actions as a chemoattractant [41], to promote cell death [42], or as a regulator of phagocytosis [43–45]. While recent studies have expanded our knowledge of how TNF signaling influences the function of immune cells in other insects [46], the role of TNF signaling in the mosquito innate immune system remains unclear. While orthologs of *Anopheles* TNF signaling components have previously been identified and support the presence of a functional pathway that influences mosquito vectorial capacity [36], direct investigations of TNF signaling have not previously been performed.

In this study, we establish integral roles of mosquito TNF signaling in mediating anti-*Plasmodium* immune responses that limit malaria parasite survival in the mosquito host. While gene expression analysis indicates that *Eiger* is induced in response to blood-feeding regardless of infection status in midgut and hemocyte tissues, downstream experiments clearly demonstrate the roles of mosquito TNF signaling in cellular immune function and immune responses that promote malaria parasite killing. Together, our data provide novel mechanistic insight into the function of TNF signaling in mosquito immune cell regulation and anti-*Plasmodium* immunity.

## Results

### Expression analyses of mosquito TNF signaling pathway components

To better understand TNF signaling in *Anopheles gambiae* we examined the expression of the TNF ligand, *Eiger*, and two TNF receptors, *Wgn* and *Grnd* (Fig 1A), across tissues (midgut, hemocytes, and carcass) and physiological conditions (naïve, blood-fed, and *Plasmodium berghei* infection). Under naïve conditions, *Eiger* displayed comparable expression levels between hemocytes and the carcass, with reduced levels of expression in the midgut (Fig 1B). A similar expression pattern was observed for *Wgn*, although higher levels of *Wgn* expression were found in carcass tissues (Fig 1C). In contrast, *Grnd* was enriched in both midgut and carcass, with the lowest expression in hemocytes (Fig 1D). Of note, *Eiger* expression was generally increased across tissues in response to blood-feeding and infection, and was significantly induced in hemocytes regardless of infection status (Fig 1B). In contrast, the different feeding conditions had little effect on *Wgn* and *Grnd* expression, although *Wgn* displayed significantly reduced levels of expression in the carcass following *P. berghei* infection (Fig 1C), suggesting the potential down-regulation of TNF signaling in the carcass following infection. While both TNF and TNF receptors (TNFRs) are influenced by posttranscriptional modifications in *Drosophila* [47–49] and other vertebrate systems, the observed patterns of *Eiger* expression support potential roles of TNF signaling in mosquito immunity and cellular immune function.

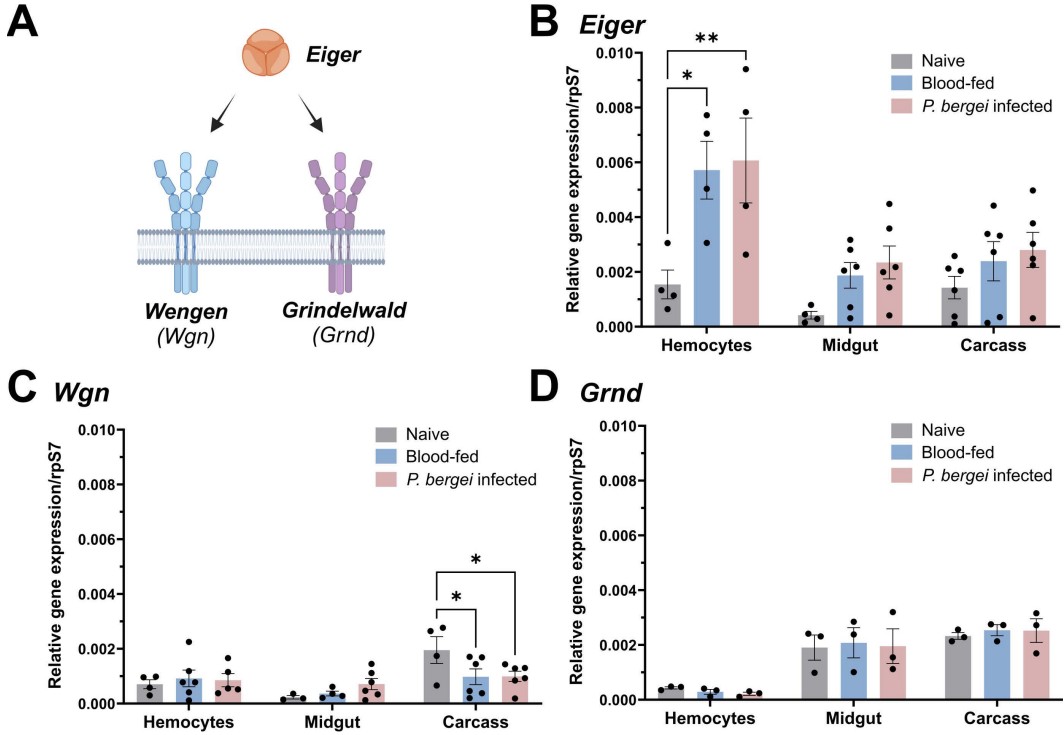

**Fig 1. Expression patterns of *Eiger*, *Wgn*, and *Grnd* in mosquitoes. (A)** Schematic representation of the mosquito TNF signaling pathway. The expression of *Eiger* **(B)**, *Wgn* **(C)**, and *Grnd* **(D)** was examined by qPCR in pooled mosquito midgut, hemolymph, and fat body samples under naive, blood-fed (24 h post-feeding) or *P. berghei*-infected (24 h post-infection) conditions. Expression data are displayed relative to rpS7 expression with bars representing the mean ± SE of three to six independent biological replicates (black dots). Each independent biological replicate consists of pooled tissues from ~10-15 individual mosquitoes for midgut and carcass samples, or hemocyte perfusions from ~30 individual mosquitoes. Data were analyzed using a two-way ANOVA with a Tukey's multiple comparisons test to determine significance. Asterisks denote significance (*$P < 0.05$, **$P < 0.01$). Summary figure created with BioRender.com.

## Influence of TNF signaling on mosquito survival

With TNF signaling components implicated in a broad range of physiological responses in other insects [38], we wanted to examine the potential that targeting these genes by RNA interference (RNAi) could potentially impact *An. gambiae* survival. After confirming that the injection of dsRNA successfully promoted the RNAi-mediated silencing of *Eiger*, *Wgn*, and *Grnd* (S1 Fig), we examined mosquito survival in each genetic background at different temperatures and following infection with the rodent malaria parasite *Plasmodium berghei* (Fig 2). When examined in insectary conditions at either 19°C or 27°C, the silencing of *Eiger*, *Wgn*, or *Grnd* had no effect on mosquito survival when compared to *GFP*-silenced control mosquitoes maintained on a 10% sucrose solution, although we did observe significant effects of temperature (19°C vs. 27°C) on mosquito survival independent of the genetic background (Fig 2A). Additional experiments were performed in which survival was examined post-silencing in mosquitoes infected with *P. berghei* (Fig 2B). While the silencing of *Eiger* or *Wgn* had no effect on survival when compared to similarly infected *GFP*-silenced control mosquitoes, *Grnd*-silencing resulted in significantly increased mosquito mortality when compared to each of the other silenced backgrounds (Fig 2B). This suggests that TNF components are not essential to mosquito survival, although under certain physiological conditions, such as infection, that TNF does contribute to mosquito fitness.

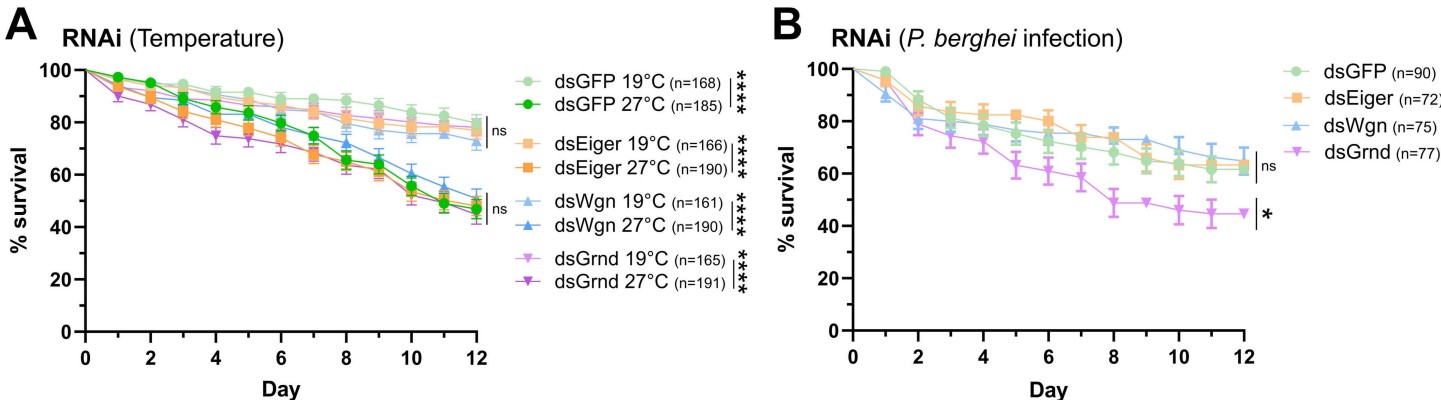

**Fig 2. Influence of TNF signaling components on mosquito survival.** *An. gambiae* survival was monitored over 12 days following the RNAi-mediated knockdown of *Eiger* (dsEiger), *Wgn* (dsWgn), *Grnd* (dsGrnd), and compared to dsGFP controls. Survival was monitored under naïve conditions at 19°C and 27°C (**A**) or was examined following *P. berghei* infection (**B**). In **A**, survival curves of each genetic background were compared for each temperature condition (significance displayed in figure) or compared within the same genetic background between temperatures (significance displayed in figure legend). Statistical analysis was performed using a Log-rank (Mantel-Cox) test, with significance displayed by asterisks: *$P < 0.05$, ****$P < 0.0001$. Error bars represent standard error of the mean (SEM). ns, not significant. n = number of individual mosquitoes in each genetic background. Data from three independent biological experiments (N = 3) are displayed for each graph.

## Mosquito TNF signaling limits *P. berghei* development

To examine the influence of TNF signaling on malaria parasite infection, we first injected adult female mosquitoes with 50 ng of recombinant human TNF-α (rTNF-α) one day before challenging with *P. berghei*. When malaria parasite numbers were evaluated 8 days post-infection (dpi), mosquitoes primed with rTNF-α significantly reduced *P. berghei* oocyst numbers and the prevalence of infection (Fig 3A). Conversely, when *Eiger* is silenced via RNAi (S1 Fig), oocyst numbers and infection prevalence were significantly increased (Fig 3B). Similarly, when the TNFRs *Wgn* and *Grnd* were silenced (S1 Fig), mosquitoes from both *Wgn-* and *Grnd*-silenced backgrounds displayed significant increases in *P. berghei* oocyst numbers (Fig 3B and 3C). Together, these findings demonstrate the importance of mosquito TNF signaling in the innate immune response against *P. berghei* and support integral roles of Eiger, Wgn, and Grnd in this process.

## Wgn and Grnd are required to promote TNF-mediated anti-*Plasmodium* immunity

To confirm that the responses limiting *P. berghei* survival following the injection of rTNF-α (Fig 3A) are mediated by mosquito TNF signaling components, we examined the influence of rTNF-α in *Wgn-* and *Grnd*-silenced backgrounds (Fig 4A). Two days after the injection of dsRNA to promote RNAi, mosquitoes from each genetic background were injected with either rTNF-α or 1X PBS as a control, then 24 hours later challenged with a *P. berghei* infection. While rTNF-α injection significantly reduced *P. berghei* oocyst numbers in the *GFP*-silenced (control) background when examined 8 dpi, the injection of rTNF-α in either the *Wgn-* or *Grnd*-silenced backgrounds had no effect (Fig 4A), suggesting that both Wgn and Grnd are required for the TNF-mediated signals that confer anti-*Plasmodium* immunity. It is also of note that we did not see significant differences in oocyst numbers between the *GFP-*, *Wgn-*, and *Grnd*-silenced backgrounds in these double-injection experiments (Fig 4A), suggesting that the additional injection post-RNAi may physically interfere with canonical immune signaling in these genetic backgrounds.

In vertebrate systems, TNF signaling can initiate distinct cellular responses based on the interactions of TNF-α with its cognate TNFR1 and TNFR2 receptors [33,34]. With both Wgn and Grnd serving as antagonists to *Plasmodium* development (Fig 3), and data suggesting that the loss of either Wgn or Grnd eliminated the effects of rTNF-α on *P. berghei* survival (Fig 4A), we wanted to examine whether TNF signaling via Wgn or Grnd produced functionally dependent or

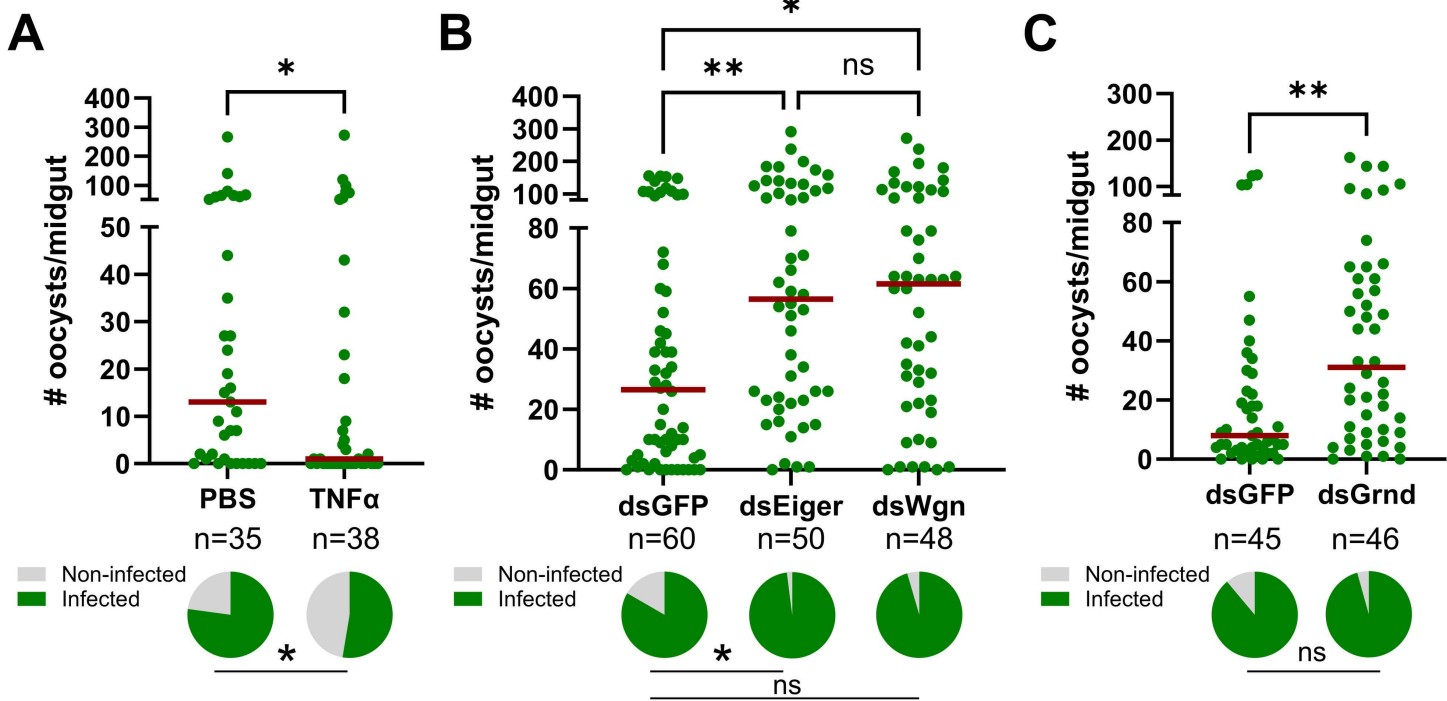

**Fig 3. TNF signaling in An. gambiae limits P. berghei survival. (A)** Adult female mosquitoes were injected with 1X PBS (control) or 50 ng of rTNF-α prior to infection with *P. berghei*. Oocyst numbers and infection prevalence were evaluated at 8 days post-infection (dpi). Additional RNAi experiments were performed to evaluate the contributions of the TNF signaling components *Eiger* and *Wgn* **(B)**, *or Grnd* **(C)** in the context of *P. berghei* infection. Oocyst numbers and infection prevalence were similarly evaluated at 8 dpi. Mosquitoes injected with dsGFP served as control in all experiments. For each graph, dots correspond to the number of oocysts identified in individual midguts, with the median represented by a red horizontal line. Infection prevalence (% infected/total) is depicted as pie charts pies below each figure. Data were combined from three or more independent experiments. Statistical significance was determined using a Mann-Whitney test or Kruskal-Wallis with a Dunn's post-test to assess oocyst numbers, while a Fisher's Exact test was performed to measure differences in infection prevalence. Asterisks denote statistical significance (* $P < 0.05$, **$P < 0.01$). ns, not significant. n = numbers of individual mosquitoes examined.

independent responses to promote parasite killing. Similar to our results in Fig 3B and 3C, the independent silencing of *Wgn* or *Grnd* resulted in increased *P. berghei* oocyst numbers (Fig 4B). However, when both *Wgn* and *Grnd* were silenced, the infection intensity did not further increase (Fig 4B). This suggests that *Wgn* and *Grnd* are functionally dependent, working together in a singular pathway to promote *P. berghei* killing, where the loss of either component abrogates TNF signaling.

## Mosquito TNF signaling modulates granulocyte and oenocytoid populations

Previous studies have demonstrated that mosquito hemocyte populations undergo significant changes in response to different physiological conditions [13–19,24,25,29] and have established their significant roles in malaria parasite killing [14–19]. Based on the upregulation of *Eiger* in perfused hemocyte samples (Fig 1) and the importance of TNF signaling associated with immune cell regulation in other systems [33,34,42,46], we wanted to explore the effects of TNF signaling on mosquito hemocyte populations.

To better understand the roles of TNF signaling in mosquito hemocytes, we first examined the expression patterns of the receptors, *Wgn* and *Grnd*, in hemocyte subtypes using previously published single-cell data for *An. gambiae* hemocyte populations [22]. While this transcriptional data suggests that *Wgn* is enriched in both granulocyte and oenocytoid

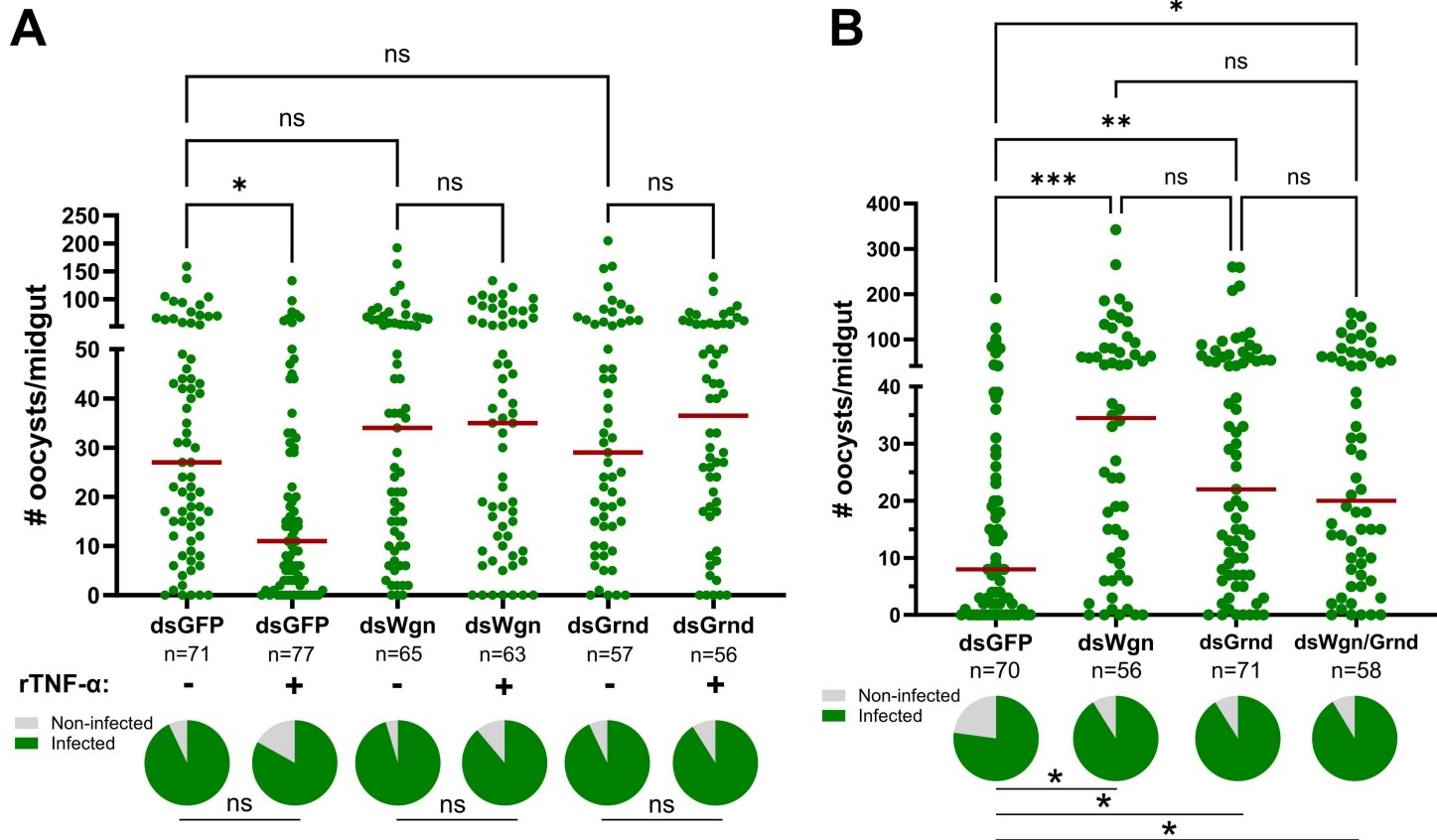

**Fig 4. TNF signaling requires the concerted function of Wgn and Grnd to promote *P. berghei* killing. (A)** RNAi experiments were performed to examine the requirement of Wgn and Grnd in rTNF-α-mediated responses that limit *P. berghei* oocyst numbers. After RNAi, mosquitoes were injected with either 1X PBS (-) or 50 ng of rTNF-α (+), then challenged with *P. berghei* one day later. **(B)** Similar RNAi experiments were performed to evaluate the combined contributions of the TNF signaling components Wgn, and Grnd in the context of *P. berghei* infection. For all experiments, oocyst numbers and infection prevalence were evaluated at 8 dpi in the specific gene-silenced backgrounds. Dots correspond to the number of oocysts identified in individual midguts, with the median represented by a red horizontal line. Infection prevalence (% infected/total) is depicted as pie charts pies below each figure. Data were combined from three or more independent experiments. Statistical significance was determined using Kruskal-Wallis with a Dunn's multiple comparison test to assess oocyst numbers, while a Fisher's Exact test was performed to measure differences in infection prevalence. Asterisks denote statistical significance (* $P < 0.05$, ** $P < 0.01$). ns, not significant; n = numbers of individual mosquitoes examined.

populations, *Grnd* is expressed only in oenocytoids (S2 Fig). These data enable a testable model in which granulocytes express *Wgn*, while oenocytoids express both *Wgn* and *Grnd* (Fig 5A). In the absence of antibodies to confirm these expression patterns, we performed additional experiments using clodronate liposomes (CLD) to deplete phagocytic granulocyte populations [17,22,50–52]. Following CLD treatment, *Wgn* displayed a significant reduction in expression (Fig 5B), while *Grnd* expression increased (Fig 5C). Together, these data demonstrate that *Wgn* is sensitive to CLD treatment, further supporting that *Wgn* is predominantly expressed in phagocytic granulocyte populations (S2 Fig). In contrast, CLD treatment did not negatively impact *Grnd* expression, supporting its expression in non-phagocytic oenocytoid populations similar to other oenocytoid-specific genes [19,22] (S2 Fig).

To determine the effects of mosquito TNF signaling on hemocyte subpopulations, we first injected mosquitoes with rTNF-α and examined the effects on granulocyte numbers. The injection of mosquitoes with either 50 ng or 200 ng of rTNF-α caused a substantial increase in the proportion of granulocytes, with the 50 ng and 200 ng concentrations able to influence granulocytes at comparable levels (Fig 5D). Based on patterns of *Wgn* and *Grnd* expression in hemocyte

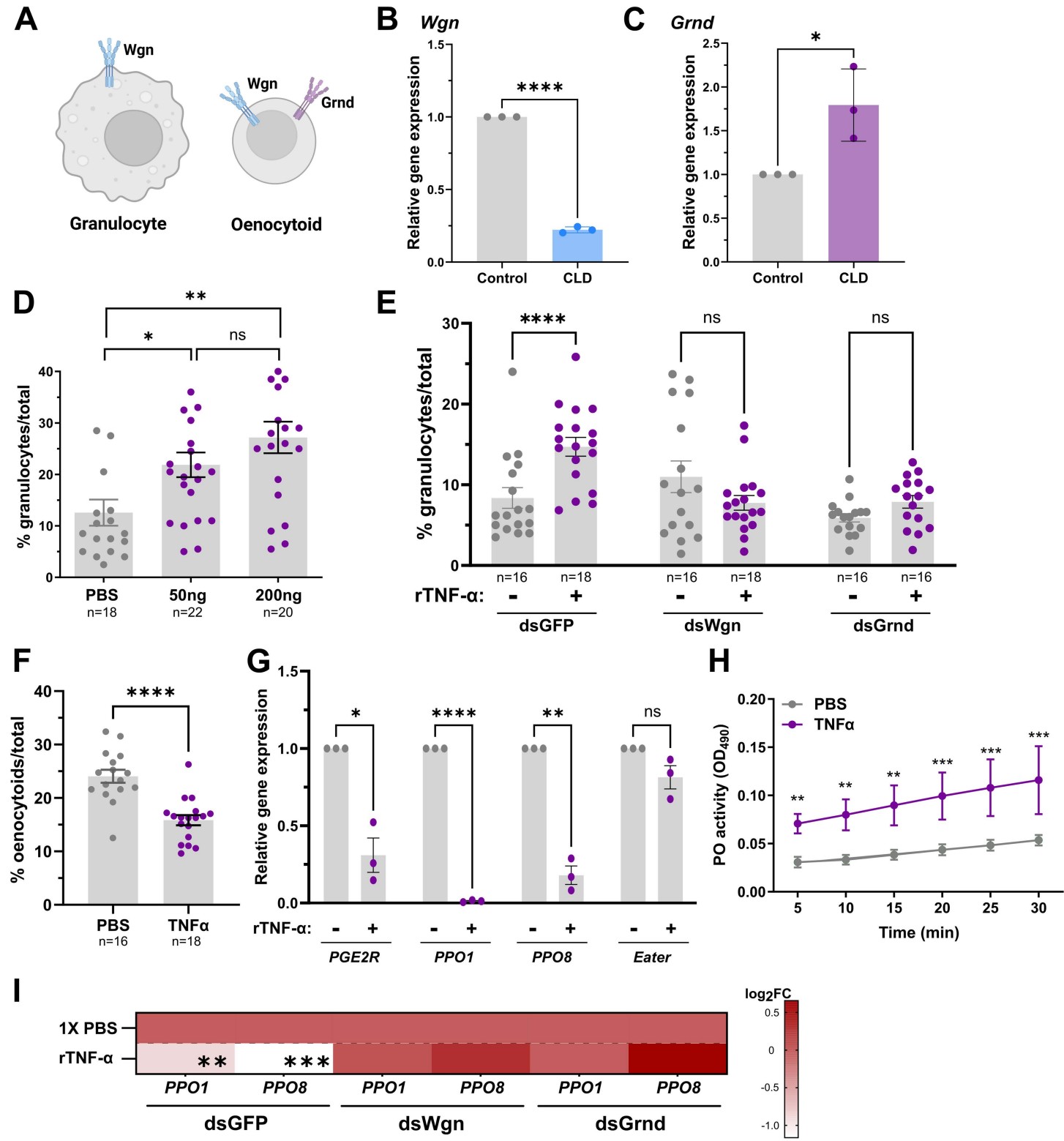

**Fig 5. TNF signaling influences granulocyte abundance and phenoloxidase activity. (A)** Previous single cell transcriptomic data [22] support the expression of *Wgn* in both granulocytes and oenocytoids, whereas *Grnd* is enriched specifically in oenocytoids. The expression levels of *Wgn* (**B**) and *Grnd* (**C**) were examined in mosquitoes treated with either clodronate liposomes (CLD) to deplete mosquito granulocytes or control liposomes to

functionally demonstrate the specificity of *Grnd* to oenocytoids using qPCR. Data from three independent experiments were analyzed by an unpaired Students' t-test. **(D)** Injection with rTNF-α (50 ng or 200 ng) increases the granulocyte proportions at 24 h post-injection compared to 1XPBS controls. Similar experiments performed in GFP-, *Wgn*-, and *Grnd*-silenced backgrounds demonstrate the importance of both *Wgn* and *Grnd* for the increase in granulocytes following rTNF-α (50 ng) injection **(E)**. For both **D** and **E**, the percentage of granulocytes of the total hemocytes are represented as mean ± SE of three independent biological replicates with statistical significance determined by Mann-Whitney to compare the effects of rTNF-α versus the 1XPBS control mosquitoes. The injection of 50 ng rTNF-α reduces the percentage of oenocytoids (of total hemocytes) **(F)** and the expression of oenocytoid specific genes (PPO1, PPO8, and PGE2R) **(G)**, which together suggest that TNF signaling promotes oenocytoid lysis. The granulocyte marker *Eater* was used as a negative control. Data from three independent experiments were analyzed by an unpaired Students' t-test. **(H)** Additional experiments were performed to determine the effects of rTNF-α on the phenoloxidase (PO)-activity of mosquito hemolymph (n = 20). Six measurements (OD490) were taken for DOPA conversion assays at 5-min intervals. Bars represent mean ± SE of three independent biological experiments with statistical significance determined with a two-way repeated-measures ANOVA followed by Sidak's multiple comparison test. **(I)** Silencing the expression of *Wgn* and *Grnd* impaired the rTNF-α induced phenotypes on PPO1 and PPO8 expression when tested in whole adult mosquitoes. Results are presented as a heatmap displaying the $\log_2$ fold change (FC) and indicate differences gene expression as measured by qPCR following treatment with rTNF-α or 1X-PBS (control). Data represent the mean fold change expression of three independent biological replicates, with significance determined using an unpaired Students' t-test. Asterisks indicate significance (* $P < 0.05$, ** $P < 0.01$, **** $P < 0.0001$). ns, not significant; n = numbers of individual mosquitoes examined. Summary figure created with BioRender.com.

subtypes (Figs 5A-C and S2), we wanted to examine the ability of rTNF-α to influence granulocyte populations in the *Wgn*- and *Grnd*-silenced backgrounds. While the injection of rTNF-α increased the percentage of granulocytes in the control *GFP*-silenced background as expected, both *Wgn*- and *Grnd*-silencing negated the effects of rTNF-α on granulocyte proportions (Fig 5E), suggesting that both *Wgn* and *Grnd* are required for the rTNF-α-mediated increase in the percentage of granulocytes. Based on evidence suggesting that *Grnd* is not expressed in granulocytes (Figs 5A-C and S2), this suggests that the effects of rTNF-α on granulocyte populations are indirectly mediated by oenocytoid function or other humoral factors.

With evidence in *Drosophila* demonstrating that TNF signaling promotes melanization through crystal cell lysis [42,44], the equivalent of mosquito oenocytoid immune cell populations, we wanted to explore if TNF signaling might similarly promote oenocytoid lysis. To address this question, we injected mosquitoes with rTNF-α and examined the percentage of oenocytoids present in circulating hemolymph. Following rTNF-α injection, we observed a significant reduction in oenocytoids (Fig 5F), suggesting that TNF signaling may similarly promote oenocytoid lysis/rupture. For this reason, we examined the expression levels of *PPO1*, *PPO8*, and *PGE2R*, genes enriched in oenocytoids [19,22], which have previously been used to illustrate oenocytoid cell rupture [19]. At 24 hrs post-injection, *PPO1*, *PPO8*, and *PGE2R* each displayed significantly reduced gene expression when compared to controls (Fig 5G), suggestive of oenocytoid lysis. In contrast, the expression of the granulocyte marker, *Eater*, remained unchanged following rTNF-α injection (Fig 5G). With the rupture of oenocytoids believed to release prophenoloxidases and other cellular contents into the hemolymph, we performed dopa conversion assays to measure hemolymph phenoloxidase (PO) activity following rTNF-α injection as an additional measure to quantify oenocytoid rupture [19]. As expected, rTNF-α treatment significantly increased hemolymph PO activity when compared to control mosquitoes (Fig 5H), providing further evidence that TNF signaling promotes oenocytoid lysis/rupture. Additional injection experiments with rTNF-α performed in *Grnd*- or *Wgn*-silenced backgrounds negated the rTNF-α-mediated down-regulation of *PPO1* and *PPO8* expression as seen in control dsGFP mosquitoes (Fig 5I), suggesting that both *Wgn* and *Grnd* are required to promote oenocytoid lysis via mosquito TNF signaling. When paired with our infection experiments demonstrating that the loss of either *Wgn* or *Grnd* impaired the TNF-mediated responses that promote parasite killing (Fig 4), these data similarly suggest that both receptors are required to initiate the TNF-mediated signals that promote cellular immune function and oenocytoid lysis.

### TNF-mediated *Plasmodium* killing is not dependent on granulocyte function

Previous studies have shown that both granulocytes and oenocytoids have central roles in anti-*Plasmodium* immunity [16,17,19,53], which given the effects of TNF signaling on granulocyte numbers and oenocytoid lysis (Fig 5), corroborate

our observations regarding the influence of mosquito TNF signaling on *Plasmodium* infection (Figs 3 and 4). Since granulocytes play pivotal roles in ookinete recognition [16,17], we examined the effects of TNF signaling on early oocyst numbers. Similar to the results presented in Fig 3A performed at 8 dpi, rTNF-α injection significantly reduced *P. berghei* early oocyst numbers by 2 dpi (Fig 6A), suggesting that TNF signaling may enhance ookinete recognition. This is further supported by the increased attachment of mosquito hemocytes to the mosquito midgut following rTNF-α injection (Fig 6B), which suggests that TNF signaling contributes to early-phase anti-*Plasmodium* killing responses. However, although TEP1 and mosquito complement are often implicated in the recognition and killing of the ookinete stage [16,17,54], the injection of rTNF-α did not significantly alter TEP1 expression (S3 Fig). To further examine the role of granulocytes in TNF-mediated parasite killing, we again employed the use of clodronate liposomes to deplete mosquito granulocyte populations [17,22,50,51]. To approach this question, we first depleted phagocytic granulocytes using clodronate liposomes, then treated mosquitoes with rTNF-α prior to challenge with *P. berghei* (Fig 6C). Before infection experiments, hemolymph was perfused and the granulocyte proportions were examined under each experimental condition to confirm granulocyte depletion. Similar to Fig 5D, the injection of rTNF-α in the control liposome background resulted in increased granulocyte numbers (Fig 6D). However, in clodronate liposome-treated mosquitoes which displayed a reduced percentage of granulocytes, the injection of rTNF-α had no effect (Fig 6D), thereby providing a methodology to examine the TNF-mediated contributions of granulocytes to anti-*Plasmodium* immunity. Similar to previous experiments (Figs 3A and 6A), the injection of rTNF-α reduced the parasite load as compared to the injection of 1X PBS in mosquitoes with a control liposome-treated background (Fig 6E). Also as expected [17], *P. berghei* oocyst numbers were significantly increased in PBS treated mosquitoes in the granulocyte-depleted background (Fig 6E). Of note, when rTNF-α injection was performed in the granulocyte-depleted background, parasite numbers were significantly reduced when compared to those treated with 1X PBS (Fig 6E). However, oocyst numbers were higher in mosquitoes injected with rTNF-α in the clodronate liposome background as compared to the control liposomes, suggesting that while granulocytes may contribute to TNF-mediated mechanisms that promote parasite killing, granulocyte depletion does not fully impair these immune responses. As a result, other TNF-mediated effects on oenocytoid immune function which influence hemolymph PO activity (Fig 5) and that have previously described roles in oocyst killing responses [17,19] may also contribute to malaria parasite killing (Fig 7). In addition, we also cannot rule out the potential that TNF signaling initiates humoral immune responses produced by the fat body or other tissues to limit parasite survival.

## Discussion

Mosquito innate immunity is an integral component of vector competence [55], therefore understanding the immune mechanisms that influence *Plasmodium* survival is essential for ongoing efforts to limit malaria transmission. While several conserved immune signaling pathways, such as Toll, IMD, and JAK/STAT, have been previously implicated in mosquito vector competence [56–59], here we provide direct evidence for the role of TNF signaling in mosquito immune function.

Our expression analysis of the mosquito TNF signaling components, Eiger, Wgn, and Grnd, suggests that TNF signaling is ubiquitous across mosquito tissues, with expression detected across midgut, hemocyte, and fat body tissues. While the expression of the receptors remained relatively constant across physiological conditions, the mosquito TNF ortholog *Eiger* was more responsive to blood-feeding or *P. berghei* infection, displaying significant induction in hemocytes. While this implies that TNF signaling is further amplified by mosquito hemocytes, potential post-translational modifications, which require cleavage to release TNF-α/Eiger in its active soluble form [60], complicate our ability to make broad sweeping conclusions from these data.

While our RNA interference (RNAi) experiments suggest that the loss of TNF signaling does not influence adult mosquito fitness under naïve conditions, these survival experiments do not exclude the potential that TNF components might still contribute to several important aspects of mosquito physiology or development. TNF signaling has been previously implicated in *Drosophila* as a master regulator of midgut homeostasis regulating lipid metabolism and intestinal stem cell

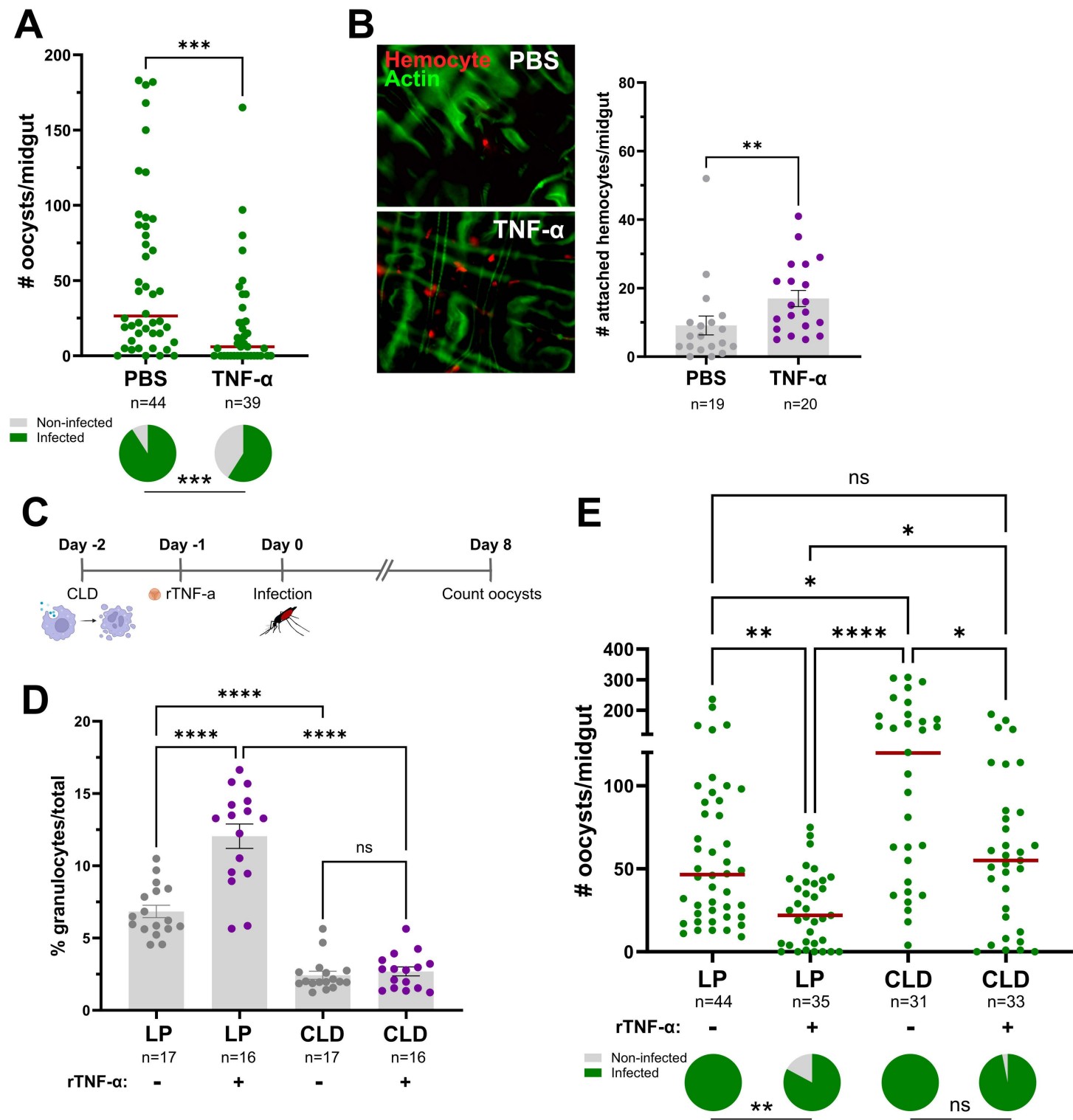

**Fig 6. TNF-mediated parasite killing targets ookinete invasion and is mediated in part by granulocyte function. (A)** Adult female mosquitoes were injected with 1XPBS (control) or 50 ng of rTNF-α prior to infection with *P. berghei*. Oocyst numbers and infection prevalence were evaluated at 2 days post infection to measure early oocyst numbers and the success of ookinete invasion. **(B)** Immunofluorescence images of DiI-stained hemocytes (red) attached to the mosquito midgut (counterstained with phalloidin, green) approximately 24 hrs post-treatment with 1xPBS (control) or 50 ng

of rTNF-α. The number of attached hemocytes was quantified for each respective treatment with the dots corresponding to the number of hemocytes attached to each individual midgut examined. To address the role of granulocytes in TNF-mediated parasite killing, mosquitoes were first injected with either clodronate liposomes (CLD) to deplete granulocytes or control liposomes (LP), then 24 hours later surviving mosquitoes were treated with 50 ng rTNF-α or 1X-PBS **(C)**. Each group was then challenged with *P. berghei* and oocyst numbers were examined on Day 8 post-infection. **(D)** Before infection, the effects of clodronate treatment on the percentage of granulocytes was examined in the presence or absence of rTNF-α to confirm our experimental approach. **(E)** Infection outcomes following granulocyte depletion and rTNF-α treatment, with oocyst numbers and infection prevalence evaluated at 8 days post infection. For both **D** and **E**, "+" denotes treatment with rTNF-α, while "-" indicates treatment with 1X-PBS. For all experiments, the dots represent the respective measurements from an individual mosquito. The red horizontal lines represent the median oocysts numbers, while infection prevalence (% infected/total) is depicted as chart pies below each figure containing infection data. Data were combined from three or more independent experiments. Statistical significance was determined using either Mann-Whitney (individual comparisons) or Kruskal-Wallis with a Dunn's multiple comparison test (multiple comparisons) to assess oocyst numbers, the number of attached hemocytes, or the percentage of granulocytes. A Fisher's Exact test was performed to measure differences in infection prevalence. Asterisks denote statistical significance (* $P < 0.05$, ** $P < 0.01$, *** $P < 0.001$, **** $P < 0.0001$). ns, not significant; n = numbers of individual mosquitoes examined. Figure created in part with BioRender.com.

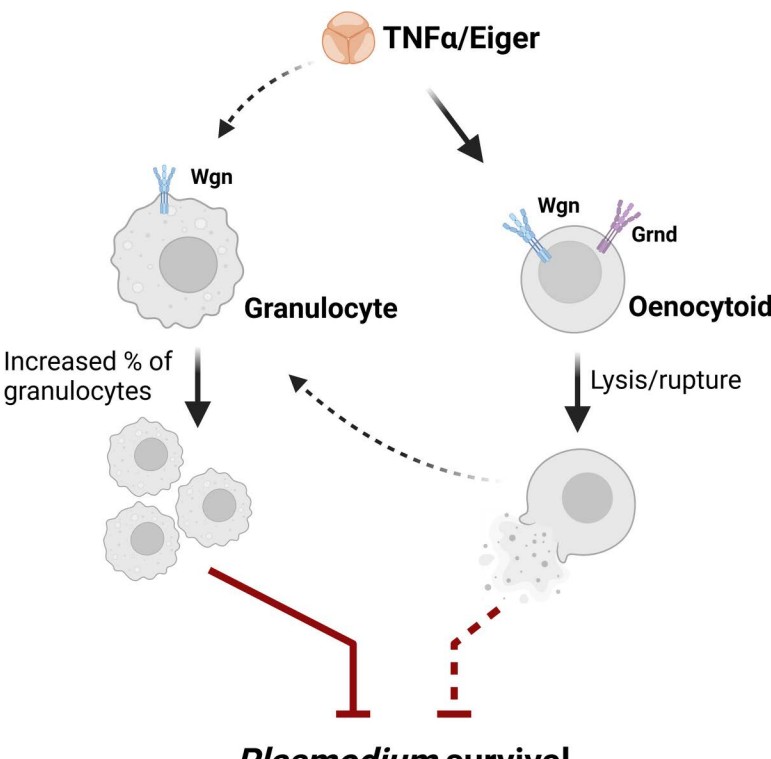

**Fig 7. Proposed model of TNF signaling on mosquito immune cells and their contributions to anti-Plasmodium immunity.** Figure created with BioRender.com.

(ISC) proliferation to maintain tissue integrity [48]. In addition, *Eiger* is expressed in *Drosophila* hemocytes in response to the midgut-derived reactive oxygen species (ROS), which ultimately triggers ISC proliferation [39]. Since mosquito blood-feeding represents a significant physiological event causing the expansion of the midgut microbiota, as well as distention and damage to the midgut epithelium, the repair of the midgut epithelium may require similar roles of intestinal stem cell differentiation [61]. This is supported by previous studies that highlight the involvement of stem cells in mosquito midgut homeostasis in response to blood-feeding, oxidative stress, and infection [62,63]. Considering the function of Eiger/Wgn signaling in *Drosophila* epithelial turnover [39,40], there is potential that the observed increase in Eiger

expression following blood-feeding and infection could similarly stimulate ISC proliferation to promote midgut homeostasis, a process that may potentially involve hemocyte function. In addition, it is unclear how the mosquito midgut microbiome could also influence this process. For example, the presence and expansion of specific microbes may contribute to dysbiosis and inflammatory-like responses that lead to Eiger activation that influence midgut damage and repair. Therefore, the observed increase in *Eiger* expression following blood feeding and infection may not only be a direct response to these physiological conditions, but also an indirect consequence of microbe-mediated modulation of TNF signaling. However, at present, direct roles of TNF signaling in mosquito midgut regeneration have yet to be determined.

Using the paired approach of rTNF-α injection and RNAi to address the potential roles of mosquito TNF signaling, we demonstrate that TNF signaling is an integral component of anti-*Plasmodium* immunity that limits *P. berghei* infection in *Anopheles gambiae*. While rTNF-α injection (and presumably overexpression of the pathway) results in reduced malaria parasite survival, loss of *Eiger*, *Wgn*, or *Grnd* prior to *P. berghei* challenge each cause an increase in *Plasmodium* oocyst numbers. While in vertebrates, TNF signaling can initiate distinct cellular responses mediated by TNFR1 and TNFR2 [64], our knockdown experiments suggest that both *Wgn* and *Grnd* mediate TNF signaling responses that are required to initiate anti-*Plasmodium* immunity. At present, it is unclear if Wgn and Grnd act as a heterodimeric receptor to promote TNF signaling, despite the lack of support from other systems. Alternatively, based on recent studies [65], Wgn and Grnd may differ in their subcellular localization and functional roles in the processing of TNF-mediated signals.

Although TNF-α is a well-established proinflammatory cytokine known for its role in regulating various aspects of macrophage function in vertebrates [66,67], our understanding of TNF signaling on insect immune cells has been limited. Previous studies have implicated Eiger/TNF in phagocytosis and host survival to pathogen infection [44–46], supporting the conservation of the TNF signaling pathway across insect taxa. Moreover, TNF signaling has been implicated in regulating phagocytic immune cell populations in solitary locusts [46] and in promoting crystal cell rupture in *Drosophila* [42]. Here, we provide evidence that TNF signaling similarly regulates mosquito immune cell function by increasing the percentage of circulating granulocyte populations and in driving oenocytoid immune cell lysis in adult female mosquitoes. While our RNA-mediated knockdowns of TNF pathway components did not influence the basal percentages of immune cell subtypes, these experiments were performed transiently in adult mosquitoes and may not fully capture potential roles of TNF signaling on hematopoiesis and immune cell maturation that may occur at earlier stages of mosquito development. As a result, further experiments are required to fully examine roles of TNF signaling on immune cell development, immune activation, and cell lysis.

As important immune sentinels, granulocytes are the primary phagocytic immune cells in the mosquito, either circulating in the open hemolymph or attached to various mosquito tissues [20]. While data support the increase in the percentage of granulocytes following rTNF-α treatment, our limited understanding of mosquito hemocyte biology and lack of genetic tools makes this a challenging phenotype to address. As a result, it remains unclear if TNF signaling via Wgn/Grnd promotes differences in cell adherence (from sessile cells to in circulation), granulocyte activation, or the differentiation of precursor cells to granulocytes. This is further complicated by the requirement of *Grnd* for the rTNF-α-mediated effects on granulocyte populations, which based on the data presented here and previous single-cell studies [22], suggest that *Grnd* is not expressed in granulocytes. As a result, we speculate that the TNF-mediated increase in granulocytes is indirect, and potentially caused by the release of other molecules resulting from oenocytoid lysis or the production of humoral factors from other tissues such as the fat body.

Similar to previous studies in *Drosophila* demonstrating the role of *Eiger* in crystal cell lysis [42], our data suggest that rTNF-α promotes the lysis/rupture of mosquito oenocytoids, the equivalent of *Drosophila* crystal cells. With evidence that *Eiger* is required for the release of prophenoloxidase [42] and melanization activity [43] in *Drosophila*, our data displaying reduced oenocytoid percentages following the injection of rTNF-α provide support for a similar mechanism in mosquitoes. These morphological observations are also supported by complementary molecular studies demonstrating that the injection of rTNF-α reduced the expression of *PPO1*, *PPO8,* and *PGE2R*, genes enriched in oenocytoid populations [22],

while promoting increased hemolymph PO activity. When placed in the context of previously published studies examining mosquito oenocytoid lysis [19], the reduced percentage of oenocytoids, reduced expression of oenocytoid markers, and increased PO activity are highly suggestive that TNF signaling regulates oenocytoid lysis/rupture. This, in turn, is believed to promote the release of prophenoloxidases that have been previously implicated in mosquito anti-bacterial [19] and anti-*Plasmodium* immunity [17,19], and potentially other immune factors that may influence mosquito innate immune function.

Previous studies indicate that both granulocytes and oenocytoids contribute to limiting malaria parasite survival in the mosquito host [16,17,19,53]. Since both immune cell subtypes display phenotypes associated with TNF signaling, we further investigated the potential roles of mosquito hemocytes in TNF-mediated parasite killing. Considering that *Plasmodium* killing in mosquitoes is multimodal, with distinct immune responses targeting either invading ookinetes or immature oocysts [15,17,54,68], we examined early (day 2) oocyst numbers as a proxy to determine the success of ookinete invasion [15,68]. Our results demonstrate a reduction in early oocyst numbers in mosquitoes injected with rTNF-α, suggesting that TNF signaling enhances *P. berghei* ookinete killing. Given that granulocytes are integral to ookinete recognition by mosquito complement [16,17], this implies that the TNF regulation of granulocyte function is central to early-phase immune responses targeting the ookinete. This is further supported by our observations of increased hemocyte attachment to the midgut following rTNF-α treatment. However, our cursory analysis of TEP1 expression following rTNF-α injection was inconclusive in implicating direct roles of mosquito complement in TNF-mediated killing responses. Moreover, through the use of clodronate liposomes to deplete phagocytic granulocyte populations [17,22,50,51], we confirm the involvement of granulocytes in TNF-mediated parasite killing. However, granulocyte depletion did not fully abrogate parasite killing following rTNF-α treatment, suggesting that additional components contribute to limiting parasite survival. It remains unclear if these TNF-mediated responses are produced by granulocyte populations not influenced by clodronate depletion [17] or if these immune responses are produced by other immune cell subtypes or tissues. Given that rTNF-α also influences oenocytoid rupture and PO activity, similar to known late-phase immune responses limiting oocyst survival via prostaglandin signaling [19], we propose a model where TNF signaling influences anti-*Plasmodium* immunity by involving both granulocyte and oenocytoid immune functions (summarized in Fig 7). Alternatively, we acknowledge the potential that TNF-mediated parasite killing responses may also be mediated in part by humoral responses produced by the fat body. Yet, due to the systemic nature of RNAi, we currently lack the genetic tools to examine the cell- or tissue-specific contributions of TNF signaling in *An. gambiae*.

In addition, an unexplored aspect of our study is the potential that blood meal-derived levels of TNF-α in host blood may be able to initiate immune signaling in the mosquito host. This is supported by previous studies which have shown that other blood meal- derived components such as insulin-like growth factor 1 (IGF1) and transforming growth factor (TGF)-beta1 are able to influence *Anopheles* mosquito vector competence to *Plasmodium* infection [69–73]. Given that acute and chronic inflammatory conditions in humans produce increased levels of TNF-α in blood [74], it remains to be seen whether certain hosts for mosquito blood meals may promote mosquito immune responses through the uptake of TNF-α in ingested blood.

While our experiments examine malaria parasite infection using the rodent malaria model, *P. berghei*, it remains to be explored if TNF pathway components similarly influence infection outcomes with the human malaria parasite, *P. falciparum*. In addition to differences in incubation temperature which may influence host physiology, there is evidence that *P. falciparum* has evolved mechanisms to evade immune recognition [75–77]. While hemocyte-mediated immune responses have been implicated in killing responses that limit both *P. berghei* and *P. falciparum* survival [15,53], it remains unclear if the TNF-mediated responses influencing cellular immune function or other humoral factors that promote *P. berghei* killing will also attenuate *P. falciparum*. As a result, further experiments are required to define the exact mechanisms by which TNF signaling influences malaria parasite survival and whether these responses are conserved across *Plasmodium* species.

In summary, our findings provide important new insights into the roles of TNF signaling in *An. gambiae,* demonstrating the effects of TNF signaling in limiting malaria parasite survival and immune cell regulation. While further study is required to fully determine the influence of TNF signaling on mosquito physiology and immune function, there is significant evidence, in addition to that provided herein, that TNF signaling is a central component that defines mosquito vectorial capacity and the susceptibility to *Plasmodium* infection in natural mosquito populations [36]. As a result, our study represents an important contribution to our understanding of the mechanisms of malaria parasite killing and the collective efforts to develop novel approaches for malaria control.

## Materials and methods

### Ethics statement

All protocols and experimental procedures regarding vertebrate animal use were approved by the Animal Care and Use Committee at Iowa State University (IACUC-21–143).

### Mosquito rearing and *plasmodium* infection

*Anopheles gambiae* mosquitoes (Keele strain) were reared at 27°C and 80% relative humidity, with a 14:10 h light: dark photoperiod cycle. Larvae were fed on commercialized fish flakes (Tetramin, Tetra), while adults were maintained on a 10% sucrose solution and fed on commercial sheep blood (Hemostat) for egg production.

Female Swiss Webster mice were used for mosquito blood-feeding and infections with a *Plasmodium berghei* (*P. berghei*) transgenic strain expressing mCherry [15,17]. For *P. berghei* infections, mice were inoculated via intraperitoneal injection using either fresh or frozen parasite stocks, then respectively monitored for the presence of exflagellating microgametes at three- or four-days post-infection as previously described [11,15]. To enable mosquito feeding, infected mice were anaesthetized via an intraperitoneal injection of ketamine:xylazine before mosquito challenge. After challenge, blood-fed mosquitoes were sorted on ice, incubated at 19°C. At either two days or eight days post-infection, individual mosquito midguts were dissected in 1X PBS, then examined by fluorescence under a compound fluorescent microscope (Nikon Eclipse 50i; Nikon) to determine parasite loads. All data are displayed in S1 Table.

### RNA extraction and gene expression analyses

Total RNA was extracted from pooled whole mosquito samples (~10 individuals) or dissected tissues (~10–15 individual mosquitoes for midgut and carcass samples, ~30 for hemocyte perfusions) using Trizol (Invitrogen, Carlsbad, CA). RNA from perfused hemolymph samples was isolated using the Direct-Zol RNA miniprep kit (Zymo Research). Two micrograms of non-hemolymph-derived or 200 ng of hemolymph-derived total RNA were used for first-strand synthesis with the RevertAid reverse transcriptase kit (Thermo Fisher Scientific). Gene expression analysis was performed with quantitative real-time PCR (qPCR) using PowerUp SYBRGreen Master Mix (Thermo Fisher Scientific) as previously described [17]. qPCR results were calculated using the $2^{-\Delta Ct}$ formula and standardized by subtracting the Ct values of the target genes from the Ct values of the internal reference, *rpS7*. All primers used in this study are summarized in S2 Table.

### Gene identification and silencing

The mosquito orthologs of known *Drosophila* TNF-α signaling components, Eiger (AGAP006771), Wengen (Wgn, AGAP000728), and Grindelwald (Grnd, AGAP008399), were identified using the OrthoDB database [78]. To address gene function, T7 primers specific to each candidate gene (S2 Table) were used to amplify DNA templates from whole female mosquito cDNA for dsRNA production and RNAi as previously [15,17]. PCR products were purified using the DNA Clean & Concentrator kit (Zymo Research) following gel electrophoresis to test for target specificity. dsRNA synthesis was performed using the MEGAscript RNAi kit (Thermo Fisher Scientific), with the concentration of the resulting dsRNA adjusted to 3 µg/µl. For RNAi, adult female mosquitoes (3–5 days old) were anesthetized on a cold block and injected intrathoracically

with 69 nl of dsRNA targeting each gene or GFP as a negative control. Co-silencing of *Wgn* and *Grnd* was accomplished by injecting mosquitoes with a solution consisting of equal parts of the dsRNA suspensions targeting each gene. Injections were performed using Nanoject III manual injector (Drummond Scientific). To assess gene-silencing, groups of ten mosquitoes were used to analyze the efficiency of dsRNA-mediated silencing at 2 days post-injection via qPCR.

## Mosquito survival experiments

To determine the effects of silencing TNF signaling components on mosquito survival, mosquitoes were injected with dsRNA targeting GFP (control), Eiger, Wgn, or Grnd as described above and maintained on a 10% sucrose solution under insectary conditions at 19°C or 27°C. Similar experiments were performed in which gene-silencing was first performed, then mosquitoes were challenged with *P. berghei* two days later. Fully engorged mosquitoes were then maintained on 10% sucrose under insectary conditions at 19°C. Mosquito survival was monitored every 24 h over a period of 12 days. All data are displayed in S1 Table.

## Injection of human recombinant TNF-α

Recombinant human TNF-a (rTNF-α; Sigma #H8916) was resuspended in 1X PBS to a stock solution of 0.72 ng/nl. Naive or dsRNA-injected mosquitoes were anesthetized and intrathoracically injected with either 69 nl of 1X PBS (control) or the stock solution of rTNF-α to administer 50 ng of protein per individual. Following injection, mosquitoes were maintained at 27°C for 24 hrs then used for downstream infection experiments or hemocyte analysis.

## Hemocyte counting

Mosquito hemolymph was collected by perfusion using an anticoagulant buffer of 60% v/v Schneider's Insect medium, 10% v/v Fetal Bovine Serum, and 30% v/v citrate buffer (98 mM NaOH, 186 mM NaCl, 1.7 mM EDTA, and 41 mM citric acid; buffer pH 4.5) as previously described [15,17,53]. For perfusions, incisions were performed on the posterior abdomen, then anticoagulant buffer (~10 µl) was injected into the thorax. Collected perfusate from individual mosquitoes were placed in a Neubauer Improved hemocytometer and observed under a light microscope (Nikon Eclipse 50i; Nikon) to distinguish hemocyte subtypes by morphology and determine the proportion of granulocytes out of the total number hemocytes in the sample. All data are displayed in S1 Table.

## Hemocyte gene expression analysis

Following mosquito injections with rTNF-α, perfused hemolymph from at least 20 mosquitoes was used for RNA extraction, cDNA synthesis, and qPCR to estimate the expression levels of hemocyte subtype gene markers (PPO1, PPO8, and PGE2R; oenocytoid-specific, and Eater; granulocyte-specific) [19,22]. All data are displayed in S1 Table.

## Characterization of hemolymph PO activity

To determine the effects of TNF-α on phenoloxidase (PO) activity, naïve mosquitoes were injected with either 1X PBS (control) or rTNF-α. At 24 h post-injection, hemolymph was perfused from 15 mosquitoes using nuclease-free water as previously described [17,19,79]. The perfusate (10 µl) was added to a 90 µl suspension of 3, 4-Dihydroxy-L-phenylalanine (L-DOPA, 4 mg/ml), then incubated at room temperature for 10 min prior to measurements of PO activity using a microplate reader at 490 nm. Samples were measured using six independent measurements at 5 min intervals. All data are displayed in S1 Table.

## *Plasmodium* infections following rTNF-α injection

After the injection of rTNF-α as described above, both control and experimental groups were challenged on a *P. berghei*-infected mouse. After selecting for blood-fed mosquitoes on ice, mosquitoes were kept at 19°C until oocyst survival was assessed at either 2- or 8- days post-infection.

PLOS Pathogens

To determine the TNF-α-mediated responses against *Plasmodium* in a granulocyte-depleted background, 3-day-old mosquitoes were first injected with either control or clodronate liposomes as previously [17]. At 24 h post-injection, each group was treated with 50 ng of rTNF-α or 1X PBS as control. Following an additional 24 h incubation, surviving mosquitoes were challenged with *P. berghei*, with oocyst numbers examined from dissected midguts at either 2- or 8-days post-infection. All data are displayed in S1 Table.

### Analysis of *Wgn* and *Grnd* expression in hemocyte subtypes

To determine the expression of TNF signaling components in mosquito hemocyte subpopulations, the expression of *Wgn* and *Grnd* was referenced with previous single-cell transcriptomic data for *An. gambiae* hemocytes [22]. Further validation was performed using methods of granulocyte depletion via clodronate liposomes as described above [17] to confirm the presence/absence of *Wgn* and *Grnd* expression in granulocyte populations using qPCR. All data are displayed in S1 Table.

### Immunofluorescent analysis of hemocytes attached to midguts

Hemocyte attachment to mosquito midguts in response to rTNF-α treatment was examined by immunofluorescence analysis as previously described [18] with slight modification. Two days after treatment with either rTNF-α or 1XPBS, mosquitoes were injected with 69 nl of 100 µM Vybrant CM-DiI cell labeling solution (ThermoFisher) and allowed to recover for 30 min at 27°C. Mosquitoes were injected with 200 nl of 16% paraformaldehyde (PFA), then the entire mosquito was immediately submerged/ incubated in a solution of 4% PFA for 40 sec prior to transfer in ice-cold 1XPBS for midgut dissection. Dissected midguts were incubated overnight in 4% PFA at 4°C for fixation. The following day, midguts were washed with ice-cold 1XPBS three times and permeabilized with 0.1% TritonX-100 for 10 minutes at room temperature. After washing three times with 1XPBS, tissues were blocked with 1% BSA in 1XPBS for 40 minutes at room temperature and stained with Phalloidin-iFluor 405 Reagent (1:400 in PBS; abcam, ab176752) for 1 hour to visualize actin filaments. Midguts were washed with 1XPBS to remove excess staining, placed on microscope slides and mounted with ProLong Diamond Antifade Mountant (ThermoFisher). Samples were imaged by fluorescence microscopy using a Zeiss Axio Imager 2 and analyzed to determine the number of hemocytes attached to individual midguts from each experimental condition.

## Supporting information

**S1 Fig. Silencing efficiency of *Eiger*, *Grnd*, and *Wgn* genes in *An. gambiae.*** Naïve adult female mosquitoes were injected with dsRNA targeting *GFP* (control), *Eiger*, *Wgn*, or *Grnd*. Two days post-injection, whole-body mosquitoes (10–15 total) were collected for RNA extraction, followed by gene expression analysis using qPCR. Data from three or more independent experiment were examined for statistical significance using an unpaired student's t-test. Asterisks indicate significance (* $P < 0.05$, **** $P < 0.0001$).
(TIF)

**S2 Fig. Expression of *Eiger*, *Wgn*, and *Grnd* in mosquito hemocyte populations.** Using previously published single-cell RNA-seq data of *An. gambiae* hemocytes [22], the expression of Eiger, Wgn, and Grnd was evaluated for each mosquito immune cell subtype. Each dot represents a cell-specific gene expression value. RPKM, Reads Per Kilobase Per Million.
(TIF)

**S3 Fig.** TEP1 expression following the injection of rTNF-α. The influence of TNF signaling on TEP1 expression was assessed in whole mosquitoes, 24 hours after the injection of 1X PBS (control) or rTNF-α injection (50 ng/mosquito).

rTNF-α. Data were collected from four independent experiments, each involving 10 female mosquitoes per replicate. Statistical analysis was performed using an unpaired Students' t-test. ns, not significant.
(TIF)

**S1 Table. Minimal dataset for submission.**
(XLSX)

**S2 Table. List of primers used for gene expression and RNAi.** Small letters indicate the T7 promoter sequence.
(PDF)

## Author contributions

**Conceptualization:** George-Rafael Samantsidis, Hyeogsun Kwon, Ryan C. Smith.

**Data curation:** George-Rafael Samantsidis, Hyeogsun Kwon, Megan Wendland, Catherine Fonder, Ryan C. Smith.

**Formal analysis:** George-Rafael Samantsidis, Hyeogsun Kwon, Catherine Fonder, Ryan C. Smith.

**Funding acquisition:** Ryan C. Smith.

**Investigation:** George-Rafael Samantsidis, Hyeogsun Kwon, Megan Wendland, Catherine Fonder, Ryan C. Smith.

**Methodology:** George-Rafael Samantsidis, Hyeogsun Kwon, Megan Wendland, Ryan C. Smith.

**Project administration:** Ryan C. Smith.

**Supervision:** Hyeogsun Kwon, Ryan C. Smith.

**Visualization:** George-Rafael Samantsidis, Hyeogsun Kwon, Ryan C. Smith.

**Writing – original draft:** George-Rafael Samantsidis, Ryan C. Smith.

**Writing – review & editing:** George-Rafael Samantsidis, Hyeogsun Kwon, Megan Wendland, Catherine Fonder, Ryan C. Smith.

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
