## [Decision Letter · Decision Letter 0]

26 Aug 2024

Dear Dr Smith,

Thank you very much for submitting your manuscript "TNF signaling mediates cellular immune function and promotes malaria parasite killing in the mosquito Anopheles gambiae" for consideration at PLOS Pathogens. As with all papers reviewed by the journal, your manuscript was reviewed by members of the editorial board and by several independent reviewers. In light of the reviews (below this email), we would like to invite the resubmission of a significantly-revised version that takes into account the reviewers' comments.

We cannot make any decision about publication until we have seen the revised manuscript and your response to the reviewers' comments. Your revised manuscript is also likely to be sent to reviewers for further evaluation.

Sincerely,

William C Gause

Academic Editor

PLOS Pathogens

Jeffrey Dvorin

Section Editor

PLOS Pathogens

Michael Malim

Editor-in-Chief

PLOS Pathogens

orcid.org/0000-0002-7699-2064

Reviewer's Responses to Questions

**Part I - Summary**

Reviewer #1: The paper of Samantsidis et. al. investigates the role of TNF in mosquito immunity to infection by Plasmodium. The paper provides new data, is mostly clear, and asks a relevant question.

The analysis falls short in a few places, employing gene expression to draw most conclusions and fails to extend the implication to mosquito survival and transmission potential.

If the above concerns can be addressed it would warrant publication.

Reviewer #2: Samantsidis GR and colleagues assessed the role of the Tumor Necrosis Factor (TNF) signaling pathway in Anopheles innate immunity and vector competence for the rodent malaria parasite. By orthology with the model system, Drosophila melanogaster, the authors identified 3 components of the TNF-α pathway: the TNF-α ligand Eiger, and its cognate receptors Wengen and Grindelwald.

The authors determined the expression profile of these TNF-α components in distinct mosquito tissues (midguts, hemocytes and carcasses) in different physiological backgrounds: naive, non-infected and P. berghei-infected blood meals. They found different expression profiles for the 3 TNF-α components with a major influence from the non-infectious blood-feeding in midguts, but P. berghei infection does not significantly influence the expression of these 3 components compared to the non-infectious background.

Then, using both ectopic addition of a human r-TNF-α and gene knockdown silencing assays, the authors show that the mosquito TNF-α pathway controls P. berghei infection and survival in Anopheles mosquitoes. Further, they showed a potential link between TNF-α mediating anti-plasmodial immunity with mosquito hemocytes and prophenoloxidase activities.

Finally, they showed a partial implication of the granulocytes in the mosquito TNF-α immune response mediating P. berghei killing.

Overall, these findings bring novelties in the complex and already well-described mosquito immune response against the early phase of the malaria parasite infection. However, the way the results and conclusions are presented brings major issues that need to be addressed by the authors.

**Part II – Major Issues: Key Experiments Required for Acceptance**

Reviewer #1: 2) Eiger appears to the only gene specifically upregulated in Figure 1, and this occurs in Hemocytes and is triggered by both infection and blood feeding. The midgut and carcass is trending towards significance and is likely underpowered at n=6. The authors should increase in the n in this experiment or combine experiments. The authors do not indicate how many times this was done, this needs to be reported.

4) There is no kinetic looking at whether TNF is playing a role in the initial establishment of infection (i.e. hours after infection), controlling parasite expansion in the mosquito, or whether mosquitoes ultimately clear the infection. The importance of a kinetic is exemplified by the board range of parasite loads observed in individual flies, some flies seem not to impacted by RNAi. Since this is an early description of the pathway an infection kinetic would be warranted.

6) On line 156 the authors need to demonstrate the efficiency of cell depletion employing this methodology. Rather than stating “which is likely the result” the authors should analyze the impact of CLD treatment on cell numbers.

11) Perhaps most surprising, the paper lacks any sort of assessment of the impact of the TNF-a pathway on mosquito survival or transmission potential. The authors should assess these parameters to warrant publication. Infection dynamics in infected insects are known determinants of subsequent transmission and disease severity (for example see Courtney et. al. PLoS Path 2017). The pathway appears to be active, impacts parasite numbers, but is it actually protective, increasing mosquito survival and does it have downstream impacts of transmission and subsequent disease.

Reviewer #2: 1. The authors used an ectopic human TNF molecule, the r-� TNF to, theoretically, promote mosquito TNF signaling. However, they bring no evidence that the r-� TNF effect is mediated by the mosquito TNF signaling pathway (i.e. via Wgn and Grnd)?

For example, is this r-� TNF effect abolished in dsWgn or dsGrnd backgrounds? Otherwise, what is the relevancy of using an ectopic human TNF molecule, if we are not sure it activates the mosquito TNF pathway? This is important because the same r-� TNF is also used to address the influence of mosquito TNF pathway on the granulocytes and the phenoloxidase activity later in the manuscript.

2. The authors mentioned that at least 3 independent replicates were performed for most figures. However, the n number of mosquitoes, representing the total number of the 3 replications, looks very weak. For example, in Figure 2A n=35; Fig.3, dsWgn, n=26. In the latter case, it would mean that an average of less than 10 mosquitoes/replication was used. 60 mosquitoes for the 3 replications would be the minimal number of mosquitoes that would lead to reliable and robust conclusions.

3. The dsWgn condition always presents a weaker number of mosquitoes compared to the dsGFP condition. Does this mean that dsWgn led to a higher mortality rate compared to dsGFP? If this is the case, Wgn would be a vital gene.

4. Figure 4 aims to illustrate the influence of the TNF pathway on mosquito hemocytes.

Figure 4D shows that ectopic addition of r-alpha TNF increases the rate of granulocytes. In Fig. 4E, the way statistical analysis was made does not properly highlight the effect of the ectopic r-alphaTNF on granulocytes. For each dsRNA condition, the authors just compared the influence of the ectopic addition of r-alpha TNF on granulocyte rate. It would be more accurate and informative to compare the dsGFP condition (with or without r-alpha TNF) with the corresponding dsWgn and dsGrnd conditions. For example, dsGFP (r-� TNF+) with dsWgn (r-alpha TNF+).

Another information could be extracted from Figure 4E: although the addition of r-� TNF increased the granulocyte rate, the TNF signaling pathway does not control the basal level of granulocytes: comparing dsGFP with dsWgn in r-� TNF(-) background or comparing dsGFP with dsGrnd in r-� TNF- background. The authors do not mention or at least discuss this.

Figure 4H: if r-alpha TNF passes through Wgn and/or Grnd, why r-alpha TNF+ background is required to get an effect on PPO1 and PPO8? I would also expect to see at least a similar tendency in the PBS background, but the authors do not explain why. It looks like there is no basal activity of the TNF pathway near on the granulocytes nor on the PPO activity, which is unlikely to be in a biological system. How could the authors explain that?

5. Figure 5: Granulocyte depletion via CLD treatment does not completely abolish the effect mediated by r-alpha TNF(+). The authors mentioned the potential implication of humoral response but without testing the hypothesis, by at least looking at the expression level of the complement-like system mediated by APL1C/LRIM1 and TEP1. As a major humoral anti P. berghei immune complex, this would show a linkage between the TNF and the complement-like system in Anopheles mosquitoes.

In addition, could the author rule out the implication of an additional population of hemocytes than granulocytes that could be involved in the midgut attachment, illustrated in Figure 5B?

Finally, although the author claims in the title and the abstract use a general term such as “Plasmodium” or “malaria parasites”, it is important to note that the experiments were only performed with the rodent malaria parasite, P. berghei but not with the human malaria parasite. Therefore, cautiousness should be considered while using general mentions such as “Plasmodium” or malaria parasite in the text, especially considering the differences in mosquito immune responses described in the literature between rodents versus human malaria parasites. This is mentioned anywhere in the text, suggesting that the authors may consider these findings as a general mechanism for all Plasmodium.spp.

**Part III – Minor Issues: Editorial and Data Presentation Modifications**

Reviewer #1: 1) In the abstract the authors state that the TNF pathway has yet to be explored in mosquito innate immunity. While the paper provides new information and there is more to learn about immunity in vector insects, this is an overstatement and the authors should give better acknowledgement to:

https://www.nature.com/articles/s41598-022-23780-y

(Particularly on lines 95-97)

3) The information provided in the Mat and Meth regarding the mosquito infections need more detail.

5) Figure 2, B and C are using the same control group and should be graphed as such.

6) On line 156 the authors begin to describe experiments employing CLD, The authors need to define what the intent of the use of CLD.

7) The logical jump from the use of previous transcriptomic data to Figure 4A on line 158 is not tenable. There has to be some representation of the actual data that leads to this conclusion.

8) The authors state “Given the absence of Grnd in granulocytes” This is gene expression, the authors need to actually show the receptor is not present.

9) The data in 4F requires better explanation, the authors that that the gene expression is associated with oenocytoids and that rTNF-a expression loss indicates lysis, but it could also indicate down-regulation, the authors need to simply quantify the eonocytoids, they are relying too heavily on gene expression. An assessment of oenocytoid apoptosis, cell death may shed further light on the impact of rTNF-a.

10) The authors need to ensure they are doing a post-test correction for comparisons between more than 2 groups.

Reviewer #2: Figure 4 title is not appropriate. I would rather propose: “The TNF signaling pathway influences the granulocyte abundance and prophenoloxidase activities.”

Discussion:

L 250-251: I found interesting the role of TNF-alpha in Drosophila intestinal stem cells and gut homeostasis. Gut homeostasis is also linked with the enteric microbiota, which, in Anopheles mosquitoes, is known as a determinant for Plasmodium infection. It would also be interesting to tackle the potential role of the mosquito TNF signaling pathway in the microbiome.

Line 300: Replace “demonstrate” with “suggest”, as correctly mentioned in Line 309. The data support this idea but do not demonstrate it.

It would be interesting to tackle the influence of a potential TNF-mediating immune priming through a blood meal on a vertebrate host carrying high levels of TNF-alpha: this physiological background is probably well described in the literature where high levels are found in patients with chronic inflammation (polyarthritis), chronic infectious diseases, pregnant women, etc…

PLOS authors have the option to publish the peer review history of their article (what does this mean? ). If published, this will include your full peer review and any attached files.

**Do you want your identity to be public for this peer review?** For information about this choice, including consent withdrawal, please see our Privacy Policy .

Reviewer #1: No

Reviewer #2: No
---

## [Decision Letter · Decision Letter 1]

26 Jun 2025

Dear Dr Smith,

We are pleased to inform you that your manuscript 'TNF signaling mediates cellular immune function and promotes malaria parasite killing in the mosquito Anopheles gambiae' has been provisionally accepted for publication in PLOS Pathogens.

Best regards,

Jeffrey D Dvorin, MD, PhD

Section Editor

PLOS Pathogens

Jeffrey Dvorin

Section Editor

PLOS Pathogens

Sumita Bhaduri-McIntosh

Editor-in-Chief

PLOS Pathogens

orcid.org/0000-0003-2946-9497

Michael Malim

Editor-in-Chief

PLOS Pathogens

orcid.org/0000-0002-7699-2064

Reviewer Comments (if any, and for reference):

Reviewer's Responses to Questions

**Part I - Summary**

Reviewer #1: The authors are investigating the role of the TNF pathway in mosquito immunity to Plasmodium infection. The authors have included new data in response to reviewer #2 that better links the pathway to rTNF-alpha treatment and impact on Plasmodium infection in the insect and this is a significant value-add to the paper.

The data clearly shows the impact of the pathway on Oocyte number in mosquitos following infection.

The paper still lacks an assessment on how this might impact the epidemiology of disease i.e., does it impact transmission potential, transmitted dose etc. and in this regard it is somewhat limited in going beyond what is already published (i.e. the work that the authors now properly incorporate in response to the initial review. The authors have included new data showing that knock down of one of the TNF pathway components (Gmd) leads to reduced mosquito survival following infection but this gene is not unregulated in response to infection, which is a bit of a disconnect in the data (see Part II below).

Reviewer #2: The authors have responded to all my requests.

This revised version of the manuscript has been improved and is, in my opinion, suitable for publication in PloS Pathogens.

**Part II – Major Issues: Key Experiments Required for Acceptance**

Reviewer #1: Reviewer #1 Comment #2

‘Kinetic’ simply refers to a time course, as indicated in the original comment the question is whether the Eiger/Gmd/Wgn response is restricting initial parasite growth, is involved in parasite killing from a peak, and/or it’s importance in ultimately clearing parasites from the mosquito. The request was that the analysis be performed in the infected group over, say 3 time points. This could be compared with Gmd RNAi treated flies for example.

The authors reference experiments that may include oocyst counts at days 2 and 8 (in the same experiment) but I cannot find this in the paper.

This reviewer does not agree with the authors assertion that ‘it is not clear as to how to provide a kinetic”’ A kinetic simply involves the same type of analysis conducted in the paper in a select number of groups over additional time points (i.e. naïve dsGFP, infected dsGFP, infected dsGmd) . The time at which the TNF pathway is impacting the infection has significant implications for how the pathway is controlling parasites in the insect. If Oocyst analysis is too time consuming is there not a qPCR based detection method for malaria parasite quantification in mosquitos?

Reviewer #1 Comment #4

At what temperature are the mosquitoes in Figure 2B maintained? It is not clear what the data at 27°C has to do with the TNF signalling pathway. It is also not clear if the infection is leading to increased mortality in the dsGFP naïve versus infected mosquitos, this should be analyzed statistically (the data is already there). There is a significant disconnect in Figure 1 and 2 as there is no induction of Gmd yet this is the RNAi treatment that impacted survival under conditions of infection, can the authors explain this? It remains unclear if those components of the TNF pathway that are activated by infection (namely Eiger) impacts transmission potential or transmitted dose. It seems to have very little impact on mosquito survival, is there any fitness cost or potential impact on the subsequent epidemiology of disease in the absence of TNF signalling. There are many assays that the authors are likely aware of that could be employed to address this, such as the dose of parasites deposited through a membrane feeder if transmission to mice is not possible.

The reference the authors were not able to find is:

https://journals.plos.org/plospathogens/article?id=10.1371/journal.ppat.1006571

Reviewer #2: No

**Part III – Minor Issues: Editorial and Data Presentation Modifications**

Reviewer #1: Reviewer #1 Comment #1

The legend to Figure 1 is much improved and better indicates that the individual data points presented are from a pool of multiple mosquitos in individual experiments. This does create a bit of an issue around how the stats are run however as not all groups were present in all experiments and that the statistics should be ‘stratified’ (so that only data from groups present in the same experiment are employed to calculate the comparative statistic).

Reviewer #1 Comment #3

When employing such methodologies it is good practice to always check the efficiency of depletion in each experiment.

The authors need to be very clear that they are using rTNFa and not the protein product of Eiger.

Reviewer #2: No

PLOS authors have the option to publish the peer review history of their article (what does this mean? ). If published, this will include your full peer review and any attached files.

**Do you want your identity to be public for this peer review?** For information about this choice, including consent withdrawal, please see our Privacy Policy .

Reviewer #1: No

Reviewer #2: No

---

## [Editor Report · Acceptance letter]

Dear Dr Smith,

We are delighted to inform you that your manuscript, "TNF signaling mediates cellular immune function and promotes malaria parasite killing in the mosquito Anopheles gambiae," has been formally accepted for publication in PLOS Pathogens.

Best regards,

Sumita Bhaduri-McIntosh

Editor-in-Chief

PLOS Pathogens

orcid.org/0000-0003-2946-9497

Michael Malim

Editor-in-Chief

PLOS Pathogens

orcid.org/0000-0002-7699-2064